# Origin of Ising magnetism in $Ca_3Co_2O_6$ unveiled by orbital imaging

Brett Leedahl[1], Martin Sundermann[1,2], Andrea Amorese[1,2], Andrea Severing[1,2], Hlynur Gretarsson[1,3], Lunyong Zhang[1], Alexander C. Komarek[1], Antoine Maignan[4], Maurits W. Haverkort [5] & Liu Hao Tjeng [1]*

The one-dimensional cobaltate $Ca_3Co_2O_6$ is an intriguing material having an unconventional magnetic structure, displaying quantum tunneling phenomena in its magnetization. Using a newly developed experimental method, $s$-core-level non-resonant inelastic x-ray scattering ($s$-NIXS), we were able to image the atomic Co $3d$ orbital that is responsible for the Ising magnetism in this system. We can directly observe that corrections to the commonly accepted ideal prismatic trigonal crystal field scheme occur in $Ca_3Co_2O_6$, and it is the complex $d_2$ orbital occupied by the sixth electron at the high-spin $Co_{trig}^{3+}$ ($d^6$) sites that generates the Ising-like behavior. The ability to directly relate the orbital occupation with the local crystal structure is essential to model the magnetic properties of this system.

[1] Max Planck Institute for Chemical Physics of Solids, Nöthnitzer Straße 40, 01187 Dresden, Germany. [2] Institute of Physics II, University of Cologne, Zülpicher Straße 77, 50937 Cologne, Germany. [3] PETRA III, Deutsches Elektronen-Synchrotron (DESY), Notkestraße 85, 22607 Hamburg, Germany. [4] Laboratoire CRISMAT, UMR 6508 CNRS-ENSICAEN, 6 bd Maréchal Juin, 14050 Caen Cedex, France. [5] Institute for Theoretical Physics, Heidelberg University, Philosophenweg 19, 69120 Heidelberg, Germany. *email: hao.tjeng@cpfs.mpg.de

Since its crystal structure was fully determined in 1996[1], $Ca_3Co_2O_6$ has garnered a large degree of attention due to its special atomic arrangements and peculiar magnetic properties. The discovery of stair-step jumps in the magnetization at regular intervals with an increasing applied field strength[2–5] is indicative of the presence of quantum tunneling phenomena[6]. This has triggered a flurry of theoretical and experimental research activities on $Ca_3Co_2O_6$[6–13] and its close derivatives[14–17].

One-dimensional chains are formed in $Ca_3Co_2O_6$ along the c-axis with alternating $CoO_6$ octahedra and trigonal prisms. In the ab-plane, the chains form a triangular-type lattice[1] (see Fig. 1). The intra-chain Co–Co coupling is ferromagnetic, while the inter-chain coupling is antiferromagnetic[2]. The unusual magnetic properties, along with its Ising-like character[4,6,18], are linked to the geometrically frustrated crystal lattice.

However, the origin or character of the presumed Ising magnetism is the subject of much discussion. To resolve this, one actually has to also simultaneously address the issue of the charge states of each of the two Co sites as well as their spin-states[7,8,19–23]. All of the previous studies relied on calculations of one type or another, with varying and conflicting outcomes, depending on what theoretical approach was used.

Herein, we make use of 3s core-level non-resonant inelastic X-ray scattering (NIXS) to determine the orbital occupation of the cobalt 3d ions. As has recently been shown, the angular dependence of the NIXS integrated intensity of the dipole-forbidden $3s \rightarrow 3d$ transition can directly map out the shape of the local orbital hole density of an ion in its ground state, without the need to do any calculational modeling of the spectral lineshapes[24]. We make use of the finding that dipole-forbidden transitions gain intensity when the NIXS experiment is carried out with large momentum transfers $\mathbf{q}$, related to the fact that then the transition operator $e^{i\mathbf{q}\cdot\mathbf{r}}$ contains such beyond-dipole terms[25]. Applying this technique to our system, we are able to identify the Co 3d orbital that is at the core of the Ising behavior. Moreover, from the experimentally determined orbital occupations we are able to unequivocally establish both the charge and spin state of each of the two types of Co ions in the crystal.

## Results

**Acquiring NIXS spectra.** Our experimental setup is illustrated in ref. [24] and a general description of the inelastic X-ray scattering method is given in ref. [26]. Incoming X-rays ($\sim 10$ keV) were scattered by the $Ca_3Co_2O_6$ single crystal with momentum transfer $\mathbf{q} = \mathbf{k}_i - \mathbf{k}_f$ and energy transfer $\hbar\omega = \hbar\omega_i - \hbar\omega_f$, where $\mathbf{k}_{i,f}$ and $\hbar\omega_{i,f}$ denote the momentum and energy of the incoming and scattered photons, respectively (see Methods). We recorded the scattered beam as a function of the sample angle $\varphi$, here defined as the angle between the fixed momentum transfer vector $\mathbf{q}$ and the c-axis of the crystal.

A collection of NIXS spectra measured at various sample angles is displayed in Fig. 2, all of which have been normalized to have the same maximum intensity at $\approx 370$ eV. The spectra show the $M_{2,3}$ edges ($3p \rightarrow 3d$) of calcium at $\approx 30$ eV and cobalt at $\approx 60$ eV and, most importantly, the dipole-forbidden $M_1$ ($3s \rightarrow 3d$) excitations of cobalt at around $\approx 100$ eV, all overlaid on a broad Compton scattering profile—caused by the inelastic scattering of photons off the electron density of the material. A close-up of the cobalt $M_1$ edge and its angular dependence on $\varphi$ is displayed in Fig. 3. Each spectrum was measured at a different sample angle $\varphi$ in the crystallographic [001] to [110] plane, with a 7.5° increment between successive spectra. We have labeled $\mathbf{q} \parallel [110]$, $\mathbf{q} \parallel [001]$, and $\mathbf{q} \parallel [\overline{1}10]$ for measurements in which the momentum transfer vector was parallel to a high symmetry direction of the crystal. Note that the Compton profile in the

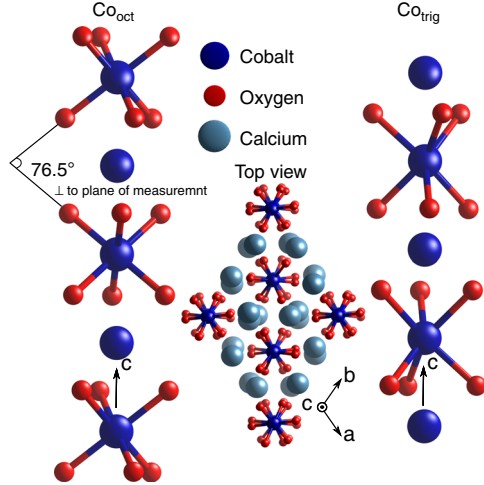

**Fig. 1** The crystal structure of $Ca_3Co_2O_6$ consists of chains of Co atoms along the c-axis, where each cobalt site alternates between trigonal and octahedral coordination with the surrounding oxygen ligands. Additionally, both trigonal and octahedral sites alternate, with each having a 45° rotation about the c-axis relative to their nearest neighbor (along the c-axis) of the same coordination. In the ab-plane (top view), the chains form a triangular-type lattice.

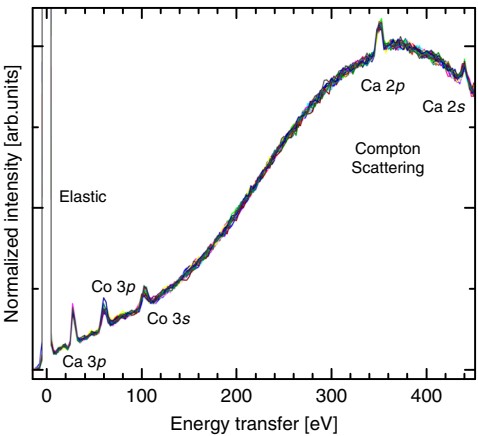

**Fig. 2** Survey spectra of scans from all sample rotation angles. The energy was scanned from the elastic peak (9.69 keV) to 450 eV above it, in order to observe the Compton scattering profile used for normalization. All calcium and cobalt absorption edges with core-level binding energies in the range can readily be seen. Of note is that dipole-forbidden Co $3s \rightarrow 3d$ transitions are visible.

narrow $M_1$ region has been subtracted using a linear background (see Methods).

The spectra in Fig. 3 taken at different sample angles in the [001] to [110] plane are composed of several peaks. We show a selection of these spectra in Fig. 4a to demonstrate that we can discern three peaks, one at 100.7 eV, one at 104.2 eV, and one in-between at 102.1 eV. Once given our final results, one will see that there is good a posteriori justification for the presence of these three peaks, and the electron configurations in Fig. 4b.

**Extracting orbitals from spectral lineshapes.** To quantitatively analyze the spectra of Fig. 3, we decomposed each spectrum into a linear combination of these three peaks, for which we are interested in their individual intensities. As illustrated in Fig. 4a, we carried out a fitting procedure in which the peaks were fit with

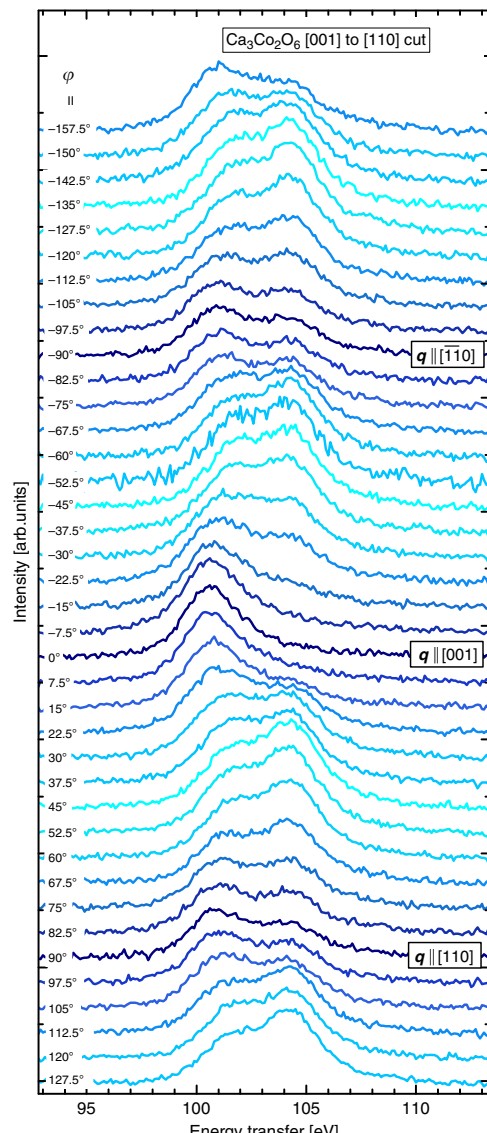

**Fig. 3** NIXS spectra at the Co $M_1$ edge. An array of spectra were recorded by rotating the sample in 7.5° increments such that **q** remained in the [110] to [001] crystallographic plane. The integrated intensity of the peaks is directly proportional to the hole density of the Co $d$-electrons in the direction of the momentum transfer **q**. The color coding is such that the darkest blue curves are spectra taken with **q** parallel to high symmetry directions of the single crystal $Ca_3Co_2O_6$.

Voigt functions to model the 1.4 eV FWHM Gaussian experimental energy broadening, and the lifetime broadening, for which a 1.8 eV FWHM Lorentzian provides the optimal fit. When implementing the fitting procedure across the array of spectra of Fig. 3, the Gaussian and Lorentzian widths of the Voigt functions were fixed, as well as the peaks' central energy positions throughout; only the intensities were free to vary to obtain the best fit to the experimental spectra.

To properly appreciate the intensity variations in the spectra one must map them to a polar plot. This is shown in Fig. 5, which displays the results from the angular scans in the [001] to [110] plane. The integrated intensity of the fitted peaks are plotted as a function of the measurement angle $\varphi$. The blue dots in Fig. 5a show the angular dependence of the intensity of the 104.2 eV peak, while the green dots in Fig. 5b show that of the summed intensities of the 100.7 and 102.1 eV peaks. Given that we are

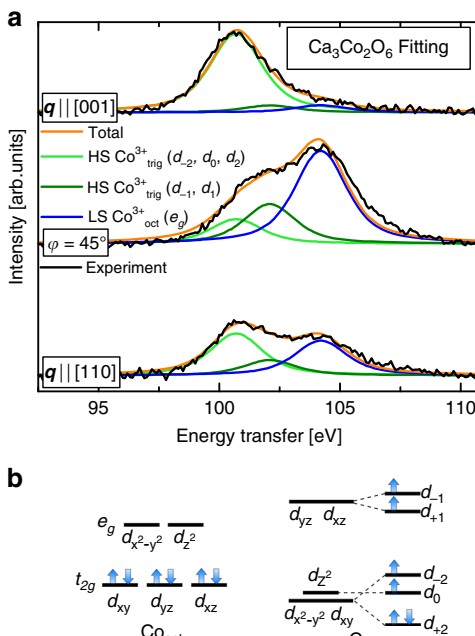

**Fig. 4** Voigt function fitting of experimental NIXS spectra. **a** The experimental spectra (black) were fit with Voigt functions that are a convolution between a 1.4 eV FWHM Gaussian function (to model the experimental resolution) and a 1.8 eV FWHM Lorentzian function (to model lifetime broadening). The component curves were integrated for each spectrum of Fig. 3, and this value is plotted on the polar plot of Fig. 5, as a function of sample angle. **b** A schematic energy level diagram for the $Co_{oct}$ site in a 3+ low-spin state shows that all the holes are in the $e_g$ orbitals, while the energy level diagram for the $Co_{trig}$ site shows the energy levels determined in this experiment. We observe that the Co is in a 3+ high-spin configuration, and that the sixth and minority-spin electron occupies the $d_2$ orbital in the ground state.

probing the hole density, we will see that there will be strong a posteriori justification at the end of this report for this particular choice of peak decomposition.

**Analysis of imaged orbitals.** To interpret the four-lobe shape of the 104.2 eV polar plot (Fig. 5a), we start with the ansatz that the $Co_{oct}$ ion is 3+ and low-spin (LS), i.e. $3d^6$ with a $t_{2g}^6$ configuration. This implies that all four holes of this ion will be in the $e_g$ shell, as depicted in Fig. 4b. To explain the asymmetries in the four-lobe shape, it is necessary to consider some of the more subtle details of the crystal structure. There are three details regarding the local coordination of $Co_{oct}$ sites that concern us: (1) along the Co–Co chains there are alternating $Co_{oct}$ sites that are rotated 45° about the $c$-axis from one another; (2) the axes along which the $d_{z^2}$ lobes lay for the two sites are oriented 13.5° off right angles from one another. Both of these are apparent in Fig. 1. Lastly, (3) we also consider the effect of the crystal field associated with the slight deviations from the perfect octahedral coordination of the $Co_{oct}$ site. However, the effect is small since the bond angles are only 3° off from the nominal 90°. In Supplementary Fig. 1 we provide a breakdown of these three contributions to the fit of the four-lobe shape, as well as details on the crystal field calculations (Supplementary Table 1).[27,28]

What follows is that due to (1), the experimental cut is a sum of a slice through the large $e_g$ lobes of one site plus the large $e_g$ lobes of the other site. These two octahedral sites along with the plane of the cut taken in the experiment are illustrated by the two lower

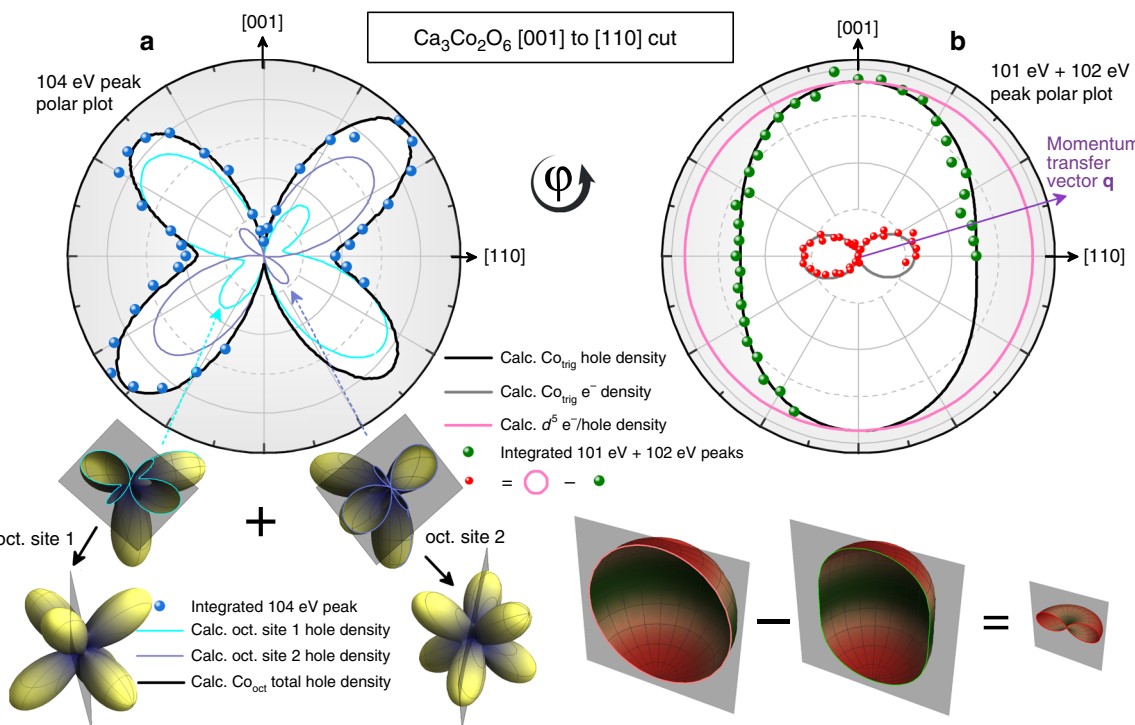

**Fig. 5** Integrated intensities of the Co $M_1$ ($3s \rightarrow 3d$) spectra of Fig. 3. The data points (derived from integrated intensities) are plotted as a function of the angle $\varphi$ between the crystallographic [001] axis and the momentum transfer vector **q**. The 3D images highlight the relevant cuts taken in the experiment in the context of the total electron/hole densities. **a** The blue dots are the result of integrating the 104.2 eV peak of Figs. 3 and 4. They agree very well with the calculated total $Co_{oct}$ hole density (black) for the candidate $d^6$ LS configuration, which is a sum of the two $Co_{oct}$ sites' hole densities (blue solid lines). The blue solid lines in the polar plot correspond to the cross-sectional cuts shown in the 3D figures beneath. **b** The calculated hole density for a $d^6$ HS configuration wherein the sixth electron occupies the complex $d_2$ orbital is shown in black, which agrees very well with our data points derived from the integrated 100.7 + 102.1 eV peaks. We can obtain the orbital wavefunction of the sixth electron (both experimentally and theoretically) by subtracting the hole density (green and black) from a reference $d^5$ circular hole density (pink). This result is shown experimentally by the red dots and theoretically with the gray solid line. The agreement reveals that the Ising-like magnetism originates from the sixth electron, which occupies the $d_2$ orbital.

3D blue-yellow shapes in Fig. 5, which show identical planes, with only the sites rotated 45° from one another as in the real crystal. In regard to (2), this means that the sum of the two sites is not done at right angles to one another, but at 76.5°. This is evident in the polar plot of Fig. 5a, which shows the contributions from both sites as solid blue lines of different shades. Finally, (3) has the subtler effect of slightly altering the relative lobe intensities. Taking into account all these details of the crystal structure, the theoretically expected curve was calculated, and is shown as a black solid line in Fig. 5a—which is a sum of the two blue curves. The near-perfect agreement between the experimental data points and the calculated curve establishes that the ansatz is correct, i.e. that the $Co_{oct}$ ions are 3+ and LS, which suggests that the magnetism must originate from the trigonal site.

We now focus our attention on the remaining intensity not yet accounted for, that is, the 100.7 and 102.1 eV peaks of Fig. 4, which must result from excitations at the $Co_{trig}$ site. Given that the $Co_{oct}$ is 3+, charge balance requires that the $Co_{trig}$ ions must also be 3+. The total integral of the 100.7 and 102.1 eV peaks leads to the conclusion that the hole density of the $Co_{trig} 3d^6$ ion has an oval shape, as shown by the green dots in Fig. 5b. To understand the meaning of this shape, consider first a high-spin (HS) $3d^5$ ion. Its hole density will be spherical since it is a half-filled shell system, where each of the five $d$-orbitals contains one electron. To represent this situation in the [001] to [110] plane of our experiment, we have added a pink circle to Fig. 5b. The difference between the circle (reference $d^5$) and the oval (measured $d^6$) then reflects the orbital that is occupied by the

extra electron that the $3d^6$ ion has in comparison to the $3d^5$ situation. This experimentally deduced shape is plotted with red dots in Fig. 5b, and physically represents the orbital shape occupied by the sixth electron.

Only now are we in a position to recognize that this is a cut through the [001] to [110] plane of the donut-like $d_2$ (or $d_{-2}$ orbital —they are identical, with only a sign difference, which indicates the direction of the magnetic moment), where the $d_m$ notation refers to the complex spherical harmonics $Y_2^m$. The expected shape of the cross-sectional cut through the $d_2$ orbital is shown as a gray solid line, which agrees very well with our experimental data points. To assist in visualizing the experimental cut through the total electron/hole density, we have plotted the 3D theoretical shapes (green–red shapes) of the corresponding cuts taken in the experiment (Fig. 5). This allows one to compare the cut in the 3D electron/hole density to the 2D data obtained in the experiment, where the highlighted outlines of the cross-sectional cuts of the 3D shapes in Fig. 5 are the expected experimental curves.

To corroborate this description and to verify that it is indeed the $d_2$ orbital that is occupied by the sixth electron, we have also obtained experimental spectra in the [100] to [010] plane (perpendicular to the Co–Co chains). Shown in Fig. 6, we can see that the expected cross-sectional cut through the $d_2$ orbital is a horizontal cut through the donut-like shape, which is circular. Our experimental data points (green dots) obtained from integrating and summing 100.7 + 102.1 eV peaks of this separate data set are adequately in agreement with the calculated hole density (black). As was previously described, the electron density was obtained by subtracting the hole density from a reference $d^5$

spherical hole density (pink). Again, the experimental red dots agree well with the calculated gray circular shape in Fig. 6. Note that Figs. 5b and 6 are different projections of the same hole density shape, as such, the radial scales have be plotted to be identical, and can be quantitatively compared.

We would like to note that by using the HS $3d^5$ sphere as reference, one may infer that we have effectively assumed that the $3d^6$ configuration of the $Co_{trig}$ is in the HS state. However, such an assumption is justifiable since the total effective magnetic moment ($\mu_{eff}^2 = 37\ \mu_B^2$/f.u.[6]), and also the saturation magnetic moment ($M_{sat} = 4.8\ \mu_B$/f.u.[6]) are larger than what an intermediate-spin or let alone a low-spin configuration could produce, taking into account that we just have firmly established that the other Co site, namely $Co_{oct}$, is non-magnetic LS $3d^6$. But it can also be simply validated on the grounds that any ansatz different than an HS configuration does not lead to a solution that reproduces the experimentally observed shape of the hole density.

We have also investigated the plausible alternative as suggested in the literature[7] in which the $Co_{oct}$ site is thought to be 4+ and LS, and the $Co_{trig}$ site to be 2+ and HS. However, we show in Supplementary Figs. 2 and 3 that the corresponding fits do not reproduce the experimental data.

## Discussion

Our finding that the $d_2$ orbital is occupied by the minority electron of the HS $Co_{trig}$ naturally explains the Ising character of the material. Since this $d_2$ orbital is the $Y_2^2$ spherical harmonic (i.e. $\ell = 2$ and $m = 2$) and the $z$-component of the orbital magnetic moment is given by $|\mu_z| = \mu_B m$, the $d_2$ orbital carries a large orbital moment of $2\,\mu_B$, and consequently generates a huge magneto-crystalline anisotropy, thereby determining the easy magnetization axis, and freezing it in the $z$-direction, i.e. along the Co–Co chain. Evidently, the crystal field potential that acts on the trigonal prismatic coordinated Co ions in $Ca_3Co_2O_6$ alters the energy levels of the $d$-orbitals in an unexpected way. The commonly accepted energy scheme that is stabilized by an ideal trigonal prismatic coordination is, from lowest to highest energy: $d_0$, $d_{\pm2}$, and $d_{\pm1}$ However, our observations contradict this scheme, and we see that it is the $d_2$ orbital that is stabilized as the lowest energy orbital.

A closer look reveals that this is in fact not trivial. Past crystal field analyses based on band structure calculations show that the $d_0$, $d_2$, and $d_{-2}$ orbitals are close in energy, but are well separated from the $d_1$, and $d_{-1}$ orbitals, as shown in Fig. 4b[7,8,19–22]. Therefore, it could have been, for example, the $d_0$ orbital that is occupied by the sixth electron, since its energy is very close to the $d_2$ orbital. The fact that small changes in energies leads to vastly different conclusions, perhaps at least partially explains why different types of past theoretical approaches have led to different explanations for the magnetism in $Ca_3Co_2O_6$[7,8,19–22]. It is therefore important that we now have an experiment by which we can image directly the occupied orbital without the need for calculations.

We are now able to retrospectively justify the identification and use of the three peaks in our experimental spectra required for producing the polar plots. First, we conclusively showed that the $Co_{oct}$ sites are in a 3+ LS state, where the six-fold degenerate $t_{2g}$ shell is completely filled, and the four-fold degenerate $e_g$ shell is completely empty (Fig. 4). Thus, the only possible excitation from the $3s$ core shell to the valence $3d$ shell is to these empty $e_g$ orbitals. This is the reason why the excitations seen in the spectra that correspond to the $Co_{oct}$ site were fit with only one peak. For the $Co_{trig}$ site, we demonstrated that it is HS $3d^6$ with the sixth and minority electron occupying the $d_2$ orbital. Therefore, the $3s$ to $3d$ excited electron can reach either one of the minority orbitals in the $d_{0/-2}$ subgroup, or the $d_{+1/-1}$ subgroup. Our best results were found via the fitting procedure when the splitting (in the

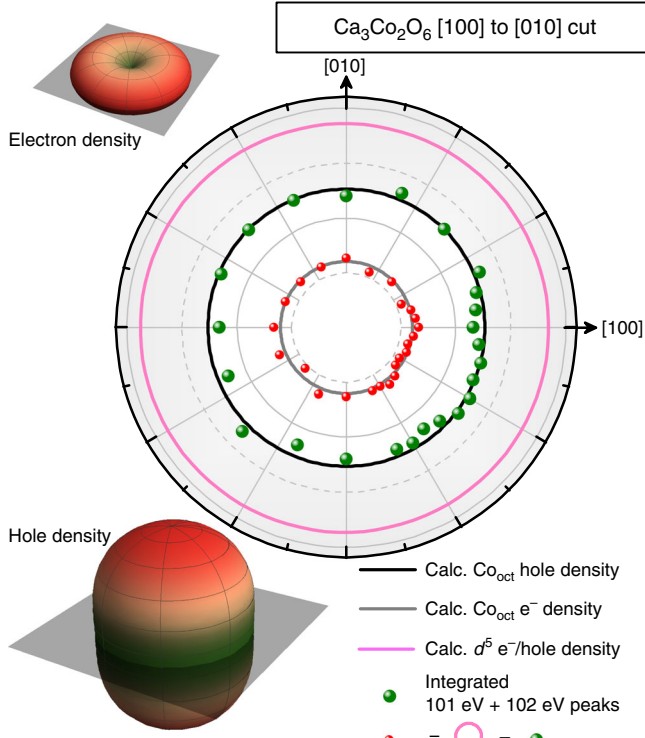

**Fig. 6** The $Co_{trig}$ site electron density cut in the $XY$-plane. The experimental/theoretical electron density (red/gray) was obtained by subtracting the experimentally/theoretically determined hole density (green/black) from a reference $d^5$ circular (spherical in 3D) hole density (pink), which is of the same radius as in Fig. 5. The insets show the calculated 3D shapes of the electorn and hole densities, along with the transparent gray planes that correspond to the experimental [100] to [011] cut. Again, we see that the circular shape of the $d_2$ orbital in this plane agrees well with our experimental data points.

presence of the $3s$ core hole) between these two subgroups was 1.4 eV. The energy separation between the peaks of the $Co_{trig}$ and $Co_{oct}$ ions reflects the chemical shift between these two sites, again, in the presence of the $3s$ core hole. Thus the existence of these peaks is retrospectively substantiated as they confirm the original hypothesis.

Utilizing the substantial intensity of the dipole-forbidden transitions available with s-NIXS, we are able to, in a straight-forward fashion, observe the electron orbitals that are active in $Ca_3Co_2O_6$. Knowledge of these orbitals is necessary for theoretical models to more reliably calculate the various inter-site exchange interactions, and in turn to quantitatively model the magnetic structure, including the intriguing magnetization tunneling phenomena. Hence, the ability to experimentally observe the relevant electron orbitals without deviation to theoretical modeling is a powerful diagnosis tool for the design of novel materials. This is especially true in situations where one would like to make use of the delicate balance of competing interactions to stabilize a particular orbital state for a desired or optimized physical property. Future studies of this nature should open new avenues for the design of materials with unusual or optimized properties.

## Methods

**Experimental details**. NIXS measurements were performed at the High-Resolution Dynamics Beamline P01 of the PETRA-III synchrotron in Hamburg, Germany. Figures in refs. [24],[26] illustrate the experimental setup. They show the incoming beam ($k_i, \omega_i$), sample, scattered beam ($k_f, \omega_f$), and the corresponding

momentum transfer vector (**q**). The energy of the X-ray photon beam incident on the sample was tuned with an Si(111) double-reflection crystal monochromator. The photons scattered from the sample were collected and energy analyzed by an array of 12 spherically bent Si(660) crystal analyzers. The analyzers are arranged in a $3 \times 4$ configuration. The energy filtering of the analyzers ($\hbar\omega_f$) was fixed at 9.69 keV; the energy loss NIXS spectra were measured by scanning the energy of the monochromator ($\hbar\omega_i$). Each analyzer signal was individually recorded by a position-sensitive custom-made LAMBDA detector. The energy calibration and resolution were ensured by pixel-wise aligning the zero energy loss feature at each measurement angle from the 12 analyzers and summing the total signal; the experimental broadening for all spectra was measured to be 1.4 eV (FWHM), as determined by the elastic peak width. However, it should be noted that the lifetime broadening is about 1.8 eV for this experiment (as determined by the remaining Lorentzian broadening unaccounted for by the experimental Gaussian broadening), resulting in a total broadening of 2.6 eV. Therefore, any attempts to significantly improve total resolution (which always comes at a large cost in flux) will be limited by the lifetime broadening inherent in the experimental lineshape.

The positioning of the analyzer array determines the momentum transfer vector and the corresponding scattering triangle, which is defined by the incident and scattered photon momentum vectors, $\mathbf{k}_i$ and $\mathbf{k}_f$, respectively. The large scattering angle ($2\theta = 155°$) and use of hard X-rays chosen for this study results in a large momentum transfer. For this experiment the magnitude of our momentum transfer vector was calculated to be $|\mathbf{q}| = (\mathbf{k}_i^2 + \mathbf{k}_f^2 - 2|\mathbf{k}_i||\mathbf{k}_f|\cos(2\theta))^{1/2} = (9.6 \pm 0.1)$ Å$^{-1}$ when averaged over all analyzers. $\mathbf{k}_f$ and $2\theta$ were kept constant by fixing the energy and the position of the analyzer array. Since the energy transfer range of interest (90–115 eV) is small with respect to the incident energy (~ 9.8 keV), variation of $\mathbf{k}_i$ during energy scanning is insignificant. This guarantees that the scattering triangle was maintained throughout the course of the experiment and that $|\mathbf{q}| \approx$ constant.

**Sample synthesis**. The single crystal $Ca_3Co_2O_6$ sample was grown using the starting materials $CaCO_3$ and $Co_3O_4$, which had a molar ratio of 4:$\frac{4}{3}$; these were then mixed with $K_2CO_3$ in a weight ratio of 1:7. The well mixed powders were then put in an aluminum crucible and heated in air to 920 °C for 9 h, and then left to dwell at this temperature for 24 h. Lastly, they were slowly cooled to 850 °C at a rate of 1 K h$^{-1}$, then to 800 °C at a rate of 3 K h$^{-1}$, and finally back to room temperature in eight more hours. Under these conditions needle-shaped crystals of several millimeters length were obtained, where the crystallographic c-axis lies along the length of the needle-shaped sample. The orientation of the crystallographic directions was ensured to within ±2° using a Laue diffractometer.

**Data treatment**. It is essential to the experiment that the normalization of the spectra is handled with care. A key observation is that the lineshape of the Compton profile does not change with angle, as can be seen from Fig. 2. This is fully consistent with the fact that the scattering geometry is kept constant while rotating the sample. On the other hand, what does vary is the experimental intensity of the Compton scattering, as this is dependent on the paths the X-rays take when entering and when scattered out of the sample. That is, the incidence angle of the X-rays relative to the sample surface varies as the sample is rotated, and therefore also the interaction depth of the X-rays and the sample. However, because the theoretical Compton scattering intensity is only proportional to the total electron density in the sample—and not the scattering geometry relative to the sample surface—we should expect that this remains constant through the experiment. Therefore, the spectra in Fig. 2 were normalized such that the Compton peak at 370 eV energy transfer was of equal intensity; and the detailed spectra (Fig. 3) in the region of the Co $M_1$ edge were likewise individually normalized using the scaling factor obtained from the Comptom profile taken at the same sample angle $\varphi$.

## Data availability
The data that support the findings of this study are available from the corresponding author upon request.

## Code availability
The code for crystal field *Quanty* calculations can be found at *quanty.org*, with specific scripts for our calculations and fitting procedure available upon request.

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

## Acknowledgements
B.L. acknowledges support from the Max Planck-University of British Columbia Centre for Quantum Materials, and M.S., A.A., and A.S. from the German funding agency DFG under Grant No SE1441-4-1. Parts of this research were carried out at PETRA-III at DESY, a member of the Helmholtz Association (HGF). We thank C. Becker, K. Höfer, and T. Mende from MPI-CPfS, and F.-U. Dill, S. Mayer, and H. C. Wille, from PETRA-III at DESY for their skillful technical support.

## Author contributions

L.H.T. and M.W.H. initiated the project; B.L., M.S., A.A., and H.G. performed the experiment; B.L. and L.H.T. analyzed the data; L.Z., A.C.K., and A.M. synthesized the samples; B.L., A.S., and L.H.T. wrote the manuscript with input from all authors.

## Competing interests

The authors declare no competing interests.
