## [Peer Review File · Nature Communications]

Reviewers' comments:

Reviewer #1 (Remarks to the Author):

The work "Origin of Ising magnetism in Ca₃Co₂O₆ unveiled by orbital imaging" by B. Leedahl et al. seeks to clarify the origin of the novel magnetic properties of Ca₃Co₂O₆ by determining the charge and spin states of its two inequivalent Co ions. [In this report, I will denote these ions as Co(I) and Co(II), where Co(I) has an octahedra coordination and Co(II) has a trigonal prismatic coordination.] The major claims of the paper are 1) that the charge and spin states of the two Co ions can be uniquely determined using s-core-level non-resonant inelastic x-ray scattering (NIXS) without calculations, and 2) that this method shows that the material's Ising behavior arises from a single unpaired electron occupying the $Y_{l=2}^{m=2}$ orbital (denoted d₂) on the Co(II) ion. The authors also stress that the stabilization of the d₂ orbital is rare in transition metal oxides.

The paper is well written, interesting, and was a pleasure to review. Overall, I find the analysis convincing but there are some places where the arguments and presentation could be improved both for specialists and for a general reader (see below). The analysis of the data relies quite heavily on the method introduced in Ref. 24 by some of the current authors. While the novelty of the method is weakened by this fact, I feel that the current work is a nice demonstration of its power. I also believe the identification of the origin of the Ising magnetism in Ca₃Co₂O₆ will be of interest not only to the specialists studying the compound but also the wider community. For these reasons, I am inclined to recommend publication but I would like the authors to first address some comments and questions listed below (in no particular order).

1) The authors claim that the charge and spin states of the two Co ions are determined without calculation. But when reading the manuscript, I get the impression that crystal field calculations were needed whose details are missing. For example, the authors state "it is also necessary to consider the effect of the crystal field, which deviates slightly from that of nominal Oh symmetry", and later "[f]inally, (3) has the more subtle effect of slightly altering the relative lobe intensities. Taking into account all these details of the crystal structure, the theoretically expected curve was calculated...". The authors need to clarify these statements and provide details. Did they do a crystal field calculation to determine the relative occupations of the orbitals? If so, please provide the details. If calculations were done with a particular package, the inputs should be provided in the supplement. (Also, "more subtle" should be "subtler".)

2) In the abstract, and later in the main text, the author state that stabilizing the d₂ orbital "is only made possible by the delicately balanced prismatic trigonal coordination." I disagree with this statement. Prismatic trigonal coordination produces crystal fields separated into three groups: a low-energy singlet state formed from the d_{z²} orbital, a mid-energy doublet formed from the d_{x²-y²} and d_{xy} orbitals, and a high-energy doublet formed from the d_{xz} and d_{yz} orbitals. This is different than the energy diagram drawn in figure 4b, which instead corresponds to the situation inferred from the data for Ca₃Co₂O₆. I believe that what authors results indicated is that corrections to the prismatic trigonal coordination crystal fields are reordering the ion's energy levels. The authors should rephrase these statements. The caption for figure 4b also needs to be modified to indicate that energy levels labeled as prismatic trigonal are in fact the reorganized levels determined in this work.

3) Related to the above, do the authors have any idea why it is the d₂ orbital and not the d₋₂ orbital that is lowest in energy? Can they distinguish between these two cases given the hole/electron density?

4) The authors use strong language such as "we conclusively showed that the Co_{oct} sites are in a 3+ LS state" at a number of points, but this is not what they have done as the work is currently presented. What they have shown is that the results are consistent with this state. To conclusively show it, they need to show that no other choice of spin and charge states can account for the data. I recognize that the number of combinations is quite large and showing/testing all combinations is not possible; however, a few examples could be provided in the supplement, which would strengthen the claims of the paper. Similarly, the authors provide good posteriori motivation for fitting the M1 peaks using three Voigt functions, and for assigning the 104.2 eV peak to the Co(I) ion and the remaining peaks to the Co(II) ion. However, I don't have a sense of how robust the conclusions are against different assignments. I suggest the authors prepare a supplement where they can show a couple of other choices and how badly alternative choices do or do not capture the experimental data.

5) A general reader might benefit from being reminded of why dipole forbidden excitations are allowed in the NIXS experiment.

Reviewer #2 (Remarks to the Author):

The authors used a unique feature of non-resonant inelastic x-ray scattering to untangle the 3d orbitals in a class of very interesting strongly correlated materials.

The data in combination with model calculations allow them to convincingly show the role of the d2 orbital in the unconventional magnetic structure and magnetism.

This approach should be of interest to the broader condensed matter physics research community in situations where theory is not sufficient to resolve closely spaced energy levels. In addition, accessibility of the inelastic x-ray scattering facilities worldwide will increase in the future as well.

One technical question I have, which should benefit the readers as well, is the limit of energy resolution of the experiment. What is the ideal energy resolution for the measurement (I assume it is limited by the lifetime of the core-hole); and what is the prospect of improving the resolution? Is there a chance that the resolution could get close of that of photoemission?

In summary, this is a good use of a novel experimental technique in answering an important question.

Reviewer #3 (Remarks to the Author):

This manuscript by Tjeng and coauthors uses a new X-ray scattering technique (s-core-level non-resonant inelastic X-ray scattering) to uncover the origin of the unusual Ising-like magnetism that has previously been observed but not explained in $\text{Ca}_3\text{Co}_2\text{O}_6$. This is possible because the dipole-forbidden 3s-3d transition has a substantial signal in the NIXS technique and it allows the cobalt 3d orbital occupations to be directly imaged, which has not been possible until now. Although details of the NIXS technique itself were reported earlier this year in Nature Physics (Ref. 24), the current manuscript represents, to the best of my knowledge, the first time that the real power of the NIXS technique has been demonstrated. I find it highly impressive that by careful design of the experiment, the orbital states of two distinct cobalt cations in the structure can be fully elucidated. Furthermore, this is done purely experimentally, without the need for complex calculations. This study not only solves a question in oxide magnetism that has been the subject of considerable debate for more than a decade, but also demonstrates the potential of NIXS to solve many other problems in condensed matter science, as orbital physics is at the root of many macroscopic physical properties. In this respect the work presented can be considered as a major advance in the broad area of condensed matter science.

The manuscript is written very clearly, in a way that is accessible to the non-expert. For technical details the reader is referred to previous reports. The experiments are very carefully designed, the results are clearly reported and interpreted, and the conclusions are sound. For the reasons mentioned above, I believe that this paper can have a high impact on any future research where orbital physics is important, and is therefore worthy of publication in Nature Communications.

There is only one statement that is unclear to me – at the top of page 11 it is stated that “the d2 orbital carries a large orbital moment of $2 \mu_B$ ”. It would help if the authors could explain where this comes from.

Our response to the report from Reviewer #1

The referee wrote:

The work “Origin of Ising magnetism in Ca₃Co₂O₆ unveiled by orbital imaging” by B. Leedahl et al. seeks to clarify the origin of the novel magnetic properties of Ca₃Co₂O₆ by determining the charge and spin states of its two inequivalent Co ions. [In this report, I will denote these ions as Co(I) and Co(II), where Co(I) has an octahedra coordination and Co(II) has a trigonal prismatic coordination.] The major claims of the paper are 1) that the charge and spin states of the two Co ions can be uniquely determined using s-core-level non-resonant inelastic x-ray scattering (NIXS) without calculations, and 2) that this method shows that the material's Ising behavior arises from a single unpaired electron occupying the $Y_{l=2}^{m=2}$ orbital (denoted d₂) on the Co(II) ion. The authors also stress that the stabilization of the d₂ orbital is rare in transition metal oxides.

The paper is well written, interesting, and was a pleasure to review. Overall, I find the analysis convincing but there are some places where the arguments and presentation could be improved both for specialists and for a general reader (see below). The analysis of the data relies quite heavily on the method introduced in Ref. 24 by some of the current authors. While the novelty of the method is weakened by this fact, I feel that the current work is a nice demonstration of its power. I also believe the identification of the origin of the Ising magnetism in Ca₃Co₂O₆ will be of interest not only to the specialists studying the compound but also the wider community. For these reasons, I am inclined to recommend publication but I would like the authors to first address some comments and questions listed below (in no particular order).

We respond:

Thank you for the thorough review and general positivity toward our work, we believe we can, and have, adequately addressed your concerns below.

The referee wrote:

- 1) The authors claim that the charge and spin states of the two Co ions are determined without calculation. But when reading the manuscript, I get the impression that crystal field calculations were needed whose details are missing. For example, the authors state “it is also necessary to consider the effect of the crystal field, which deviates slightly from that of nominal Oh symmetry”, and later “[f]inally, (3) has the more subtle effect of slightly altering the relative lobe intensities. Taking into account all these details of the crystal structure, the theoretically expected curve was calculated...”. The authors need to clarify these statements and provide details. Did they do a crystal field calculation to determine the relative occupations of the orbitals? If so, please provide the details. If calculations were done with a particular package, the inputs should be provided in the supplement. (Also, “more subtle” should be “subtler”.)*

We respond:

We sympathize with the referee's confusion here, and agree that clarification is required. Firstly, we misspoke in claiming that it is *necessary* to consider the crystal field. It is not *necessary* in the strict sense of the word. We can get already a very good agreement by neglecting the crystal field altogether. However, we know from the crystal structure that there are tiny distortions in the octahedra. We thus feel obliged to investigate the effect of the corresponding crystal fields. We found that the inclusion of those small crystal fields does not alter our conclusions, and in fact, it provides just a slightly better agreement.

To demonstrate this clearly to the reader, we have included the following figure and caption to the Supplemental Material in which a comparison of the calculation with and without crystal field is shown.

— fit with only 45° rotation
 — fit with 45° rotation & 13.5° tilt
 — fit with complete crystal field calculation
 • Integrated 104 eV peak

FIG S1. The experimental data points (blue) have been plotted along with three fits to show the effect of the crystal structure. The green curve is the fit including only that the two low-spin Co^{3+} sites

alternatingly rotated 45 degrees about the c -axis from one another. With only this consideration, the overall shape and orientation of the experimental data set are reasonably reproduced. Next, the red curve is the fit which also includes the 13.5 degree tilt off perpendicular from one another (see manuscript Fig. 1). The agreement with the experiment is now very good. The black curve (as displayed in Fig.

5a) is the fit which also includes the effect of the crystal fields

In the interest of full disclosure and reproducibility of crystal field calculations as requested by the referee, the following table and caption have also been added to the Supplemental Material:

$A_{k,m}$	
$A_{0,0}$	0.00000
$A_{2,-2}$	$0.30878+i*(0.06982)$
$A_{2,-1}$	$-0.58775+i*(0.20583)$
$A_{2,0}$	$0.13437+i*(-0.00000)$
$A_{2,1}$	$0.58775+i*(0.20583)$
$A_{2,2}$	$0.30878+i*(-0.06982)$
$A_{4,-4}$	$2.33651+i*(-0.00709)$
$A_{4,-3}$	$0.30052+i*(0.35238)$
$A_{4,-2}$	$-0.00191+i*(0.31177)$
$A_{4,-1}$	$-0.06235+i*(0.16160)$
$A_{4,0}$	$3.91300+i*(0.00000)$
$A_{4,1}$	$0.06235+i*(0.16160)$
$A_{4,2}$	$-0.00191+i*(-0.31177)$
$A_{4,3}$	$-0.30052+i*(0.35238)$
$A_{4,4}$	$2.33651+i*(0.00709)$

TABLE S1. The potential due to the effect of the crystal field in a point charge model was calculated using the code in Ref. [1], wherein the only input is the crystal structure, following the $A_{k,m}$ nomenclature of Ref. [1]. These coefficients were then used to calculate a density matrix using the quantum many body script language *Quanty*[2]. The density matrix contains coefficients for calculating the 3D charge and hole densities using the *Mathematica* toolboxes provided on *quanty.org*.

- [1] M.W. Haverkort. *Spin and orbital degrees of freedom in transition metal oxides and oxide thin films studied by soft x-ray absorption spectroscopy*. PhD thesis, Universitat Koeln. pg 161. (2005)
 [2] M.W. Haverkort, M. Zwierzcki, and O.K. Andersen. *Multiplet ligand-field theory using Wannier orbitals*. Phys. Rev. B 85, 165113 (2012).

We have amended the text on lines 106 to 111 to be more transparent to the reader, so as there is no doubt as to how the results were obtained:

Lastly, (3) we also consider the effect of the crystal field associated with the slight deviations from the perfect octahedral coordination of the Co_{Oct} site. However, the effect is small since the bond angles are only 3 degrees off from the nominal 90 degrees. In the *Supplemental Material* we provide a breakdown of these three contributions to the fit of the four-lobe shape, as well as details on the crystal field calculations.

The referee wrote:

- 2) *In the abstract, and later in the main text, the author state that stabilizing the d_2 orbital “is only made possible by the delicately balanced prismatic trigonal coordination.” I disagree with this statement. Prismatic trigonal coordination produces crystal fields separated into three groups: a low-energy singlet state formed from the d_{z^2} orbital, a mid-energy doublet formed from the $d_{x^2-y^2}$ and d_{xy} orbitals, and a high-energy doublet formed from the d_{xz} and d_{yz} orbitals. This is different than the energy diagram drawn in figure 4b, which instead corresponds to the situation inferred from the data for $\text{Ca}_3\text{Co}_2\text{O}_6$. I believe that what authors results indicated is that corrections to the prismatic trigonal coordination crystal fields are reordering the ion’s energy levels. The authors should rephrase these statements. The caption for figure 4b also needs to be modified to indicate that energy levels labeled as prismatic trigonal are in fact the reorganized levels determined in this work.*

We respond:

We accept the remarks of the referee. The following changes to the manuscript have been made:

The abstract, on lines 21 to 23 now reads:

We can directly observe that corrections to the commonly accepted ideal prismatic trigonal crystal field scheme occur in $\text{Ca}_3\text{Co}_2\text{O}_6$, and it is the complex d_2 orbital occupied by the sixth electron at the high-spin $\text{Co}^{3+}_{\text{trig}}(d^6)$ sites that generates the Ising-like behavior.

The main text on lines 180 to 184 now read:

Evidently, the crystal field potential that acts on the trigonal prismatic coordinated Co ions in $\text{Ca}_3\text{Co}_2\text{O}_6$ alters the energy levels of the d-orbitals in an unexpected way. The commonly accepted energy scheme that is stabilized by an ideal trigonal prismatic coordination is, from lowest to highest energy: d_0 , $d_{+/-2}$, and $d_{+/-1}$. However, our observations contradict this scheme, and we see that it is the d_2 orbital that is stabilized as the lowest energy orbital.

The caption for Figure 4b now states:

...while the energy level diagram for the Co_{trig} site shows the energy levels determined *in this experiment*. We observe that the Co is in a 3+ high spin configuration, and that the sixth and minority-spin electron occupies the d_2 orbital in the ground state.

The referee wrote:

- 3) *Related to the above, do the authors have any idea why it is the d_2 orbital and not the d_{-2} orbital that is lowest in energy? Can they distinguish between these two cases given the hole/electron density?*

We respond:

This is a valid point, and we agree that it should be addressed. The answer is no, we cannot distinguish them. The difference between the two is only that the magnetic moment points in the opposite direction. That is, an occupied d_2 orbital would produce a magnetic moment along the c -axis, and the d_{-2} would produce a magnetic moment along the $-c$ -axis.

This information is now conveyed to the reader on the amended lines 138 to 141:

Only now are we in a position to recognize that this is a cut through the [001] to [110] plane of the donut-like d_2 (or d_{-2} orbital—they are identical, with only a sign difference, which indicates the direction of the magnetic moment), where the d_m notation refers to the complex spherical harmonics Y^m_2

The referee wrote:

4) The authors use strong language such as “we conclusively showed that the Co_{oct} sites are in a 3+ LS state” at a number of points, but this is not what they have done as the work is currently presented. What they have shown is that the results are consistent with this state. To conclusively show it, they need to show that no other choice of spin and charge states can account for the data. I recognize that the number of combinations is quite large and showing/testing all combinations is not possible; however, a few examples could be provided in the supplement, which would strengthen the claims of the paper. Similarly, the authors provide good posteriori motivation for fitting the M1 peaks using three Voigt functions, and for assigning the 104.2 eV peak to the Co(I) ion and the remaining peaks to the Co(II) ion. However, I don't have a sense of how robust the conclusions are against different assignments. I suggest the authors prepare a supplement where they can show a couple of other choices and how badly alternative choices do or do not capture the experimental data.

We respond:

We agree with the reviewer, the wording is strong, and we have not demonstrated all possible alternatives. However, we believe we can convince the reviewer by providing alternative (plausible) solutions, and stand by our word choice in this case.

The primary candidate that could be considered as an alternative is the one often suggested in past publications, such as this one: <https://doi.org/10.1103/PhysRevLett.91.186404>. This is a Co⁴⁺_{oct} LS configuration, i.e. four e_g holes and one t_{2g} hole. In the below plot (which is also included in the Supplemental Material) we have calculated the expected curve (red) if this were the case. This calculation includes a sum of the two octahedral sites, taking all the careful considerations we did for the “best fit” spectrum (black) in order to achieve the best possible agreement. Clearly, the red fit is not as good as the black.

FIG. S2. To convincingly show that our best solutions in Fig. 5 of the main manuscript are indeed a better description of the data than any alternative fit, we have calculated a commonly cited alternative solution for Co_{oct} site if it were Co⁴⁺ and low spin ($t_{2g}^5 e_g^0$), which produces a lower quality fit to the experimental data points.

As was done in the manuscript, for charge balance, this Co⁴⁺_{oct} LS scenario would imply that the trigonal Co site is 2+. The calculated hole density for this configuration is also displayed in the Supplemental Material:

FIG. S3. The Co_{trig} configuration that corresponds to the alternative choice for the Co_{oct} site shown in Fig. S2. No possible planar cut through this shape can reproduce the oval-like shape of the experimental data points shown in Fig. 5b of the main manuscript.

Remembering that our experimental data points resembled an oval shape, one can imagine that there is no possible cross-sectional cut through this shape that would produce an oval.

To communicate to the reader that we have indeed considered alternative solutions, we have added the following text to the main manuscript on lines 170 to 173

We have also investigated the plausible alternative as suggested in the literature [7] in which the Co_{oct} site is thought to be 4+ and LS, and the Co_{trig} site to be 2+ and HS. However, we have shown in the *Supplemental Material* that the corresponding fits do not reproduce the experimental data.

The referee wrote:

- 5) *A general reader might benefit from being reminded of why dipole forbidden excitations are allowed in the NIXS experiment.*

We respond:

We have added a brief description to the manuscript with also a new Ref. [25] to refer the reader to a more detailed description. The following has been added to lines 47 to 50:

We make use of the finding that dipole forbidden transitions gain intensity when the NIXS experiment is carried out with large momentum transfers \mathbf{q} , related to the fact that then the transition operator $e^{i\mathbf{q}\cdot\mathbf{r}}$ contains such beyond-dipole terms [25].

[25] Haverkort, M. W., Tanaka, A., Tjeng, L. H. & Sawatzky, G. A. *Nonresonant inelastic x-ray 298 scattering involving excitonic excitations: The examples of NiO and CoO*. Phys. Rev. Lett. 299 99, 257401 (2007). <https://link.aps.org/doi/10.1103/PhysRevLett.99.257401>.

Our response to the report from Reviewer #2

The referee wrote:

The authors used a unique feature of non-resonant inelastic x-ray scattering to untangle the 3d orbitals in a class of very interesting strongly correlated materials.

The data in combination with model calculations allow them to convincingly show the role of the d2 orbital in the unconventional magnetic structure and magnetism.

This approach should be of interest to the broader condensed matter physics research community in situations where theory is not sufficient to resolve closely spaced energy levels. In addition, accessibility of the inelastic x-ray scattering facilities worldwide will increase in the future as well.

In summary, this is a good use of a novel experimental technique in answering an important question.

We respond:

The authors would like to thank you for recommending our manuscript for publication, and noting the capability of the method to solve a real physical problem that is not easily answered with other techniques.

The referee wrote:

One technical question I have, which should benefit the readers as well, is the limit of energy resolution of the experiment. What is the ideal energy resolution for the measurement (I assume it is limited by the lifetime of the core-hole); and what is the prospect of improving the resolution? Is there a chance that the resolution could get close of that of photoemission?

We respond:

Our experimental resolution was 1.4 eV, this is noted on line 320 of the manuscript. We can improve our experimental resolution at the P01 beamline at DESY by changing from the Si(111) reflection of the double crystal monochromator to the Si(311) reflection. This configuration will provide a 0.7 eV experimental broadening, although this comes at loss of 80% of the photons.

Let us elaborate: included in our peak fitting procedure is the convolution with a 1.8 eV (FWHM) Lorentzian to simulate the lifetime broadening, for a total broadening of 2.6 eV. If we were to use the Si(311) configuration, the Gaussian contribution to the broadening would be reduced to 0.7 eV, which along with the lifetime broadening would provide a total broadening of 2.1 eV.

In this experiment an improvement from 2.6 eV to 2.1 eV broadening will not provide us with any new information since we are just integrating a few very broad peaks, and therefore is not worth the cost in flux that accompanies the resolution improvement.

Lines 335 to 339 have been updated to read:

However, it should be noted that the lifetime broadening is about 1.8 eV for this experiment, giving a total broadening of 2.6 eV. Therefore, any attempts to significantly improve total resolution (which always comes at a large cost in flux) will be limited by the lifetime broadening inherent in the experimental lineshape.

Our response to the report from Reviewer #3

The referee wrote:

This manuscript by Tjeng and coauthors uses a new X-ray scattering technique (s-core-level non-resonant inelastic X-ray scattering) to uncover the origin of the unusual Ising-like magnetism that has previously been observed but not explained in Ca₃Co₂O₆. This is possible because the dipole-forbidden 3s-3d transition has a substantial signal in the NIXS technique and it allows the cobalt 3d orbital occupations to be directly imaged, which has not been possible until now. Although details of the NIXS technique itself were reported earlier this year in Nature Physics (Ref. 24), the current manuscript represents, to the best of my knowledge, the first time that the real power of the NIXS technique has been demonstrated. I find it highly impressive that by careful design of the experiment, the orbital states of two distinct cobalt cations in the structure can be fully elucidated. Furthermore, this is done purely experimentally, without the need for complex calculations. This study not only solves a question in oxide magnetism that has been the subject of considerable debate for more than a decade, but also demonstrates the potential of NIXS to solve many other problems in condensed matter science, as orbital physics is at the root of many macroscopic physical properties. In this respect the work presented can be considered as a major advance in the broad area of condensed matter science.

The manuscript is written very clearly, in a way that is accessible to the non-expert. For technical details the reader is referred to previous reports. The experiments are very carefully designed, the results are clearly reported and interpreted, and the conclusions are sound. For the reasons mentioned above, I believe that this paper can have a high impact on any future research where orbital physics is important, and is therefore worthy of publication in Nature Communications.

We respond:

We would like to thank you for the positive review and appreciation of the power of this innovative technique and its ability to solve outstanding questions purely experimentally.

The referee wrote:

There is only one statement that is unclear to me – at the top of page 11 it is stated that “the d₂ orbital carries a large orbital moment of 2 μ_B”. It would help if the authors could explain where this comes from.

We respond:

The theoretical value of 2μ_B arises from the fact that it is the Y₂^m = Y₂² spherical harmonic that the electron occupies as its orbital; l = 2 for d-electrons. Then, the magnitude of the z-component of the orbital magnetic moment is:

$$\mu_z = - \left(\frac{e}{2m_e} \right) L_z = - \left(\frac{e}{2m_e} \right) m\hbar = -\mu_B m.$$

where the negative sign arises from the fact that it is due to negatively charged electrons.

And since m = 2 for this orbital, the magnitude of our orbital moment is 2μ_B along the Co-Co chains, similar to what has been measured.

A briefly summarized version of this has been added to the manuscript on lines 175 to 177:

Since this d₂ orbital is the Y₂² spherical harmonic (i.e. l = 2 and m = 2) and the z-component of the orbital magnetic moment is given by |μ_z| = μ_Bm, the d₂ orbital carries a large orbital moment of 2μ_B...

REVIEWERS' COMMENTS:

Reviewer #1 (Remarks to the Author):

I have examined the reply from the authors and the revised manuscript. At this time, the authors have addressed my comments to my satisfaction and I recommend that the paper be accepted for publication in its current form. I also would like to thank the authors for taking the time to address my comments and for expanding the supplementary materials.

Steve Johnston

Reviewer #2 (Remarks to the Author):

The authors didn't quite answer my question about energy resolution. To be more clear, is the 10.8 eV lifetime broadening just a fit to the experimental data or measured independently, or from literature? If it is only a fit to the data, then improving experimental energy resolution might help.

Reviewer #3 (Remarks to the Author):

The authors have clearly answered my previous question regarding the orbital moment associated with the d₂ orbital. The manuscript has also been further strengthened in response to the comments of the other referees. In particular, additional crystal field calculations have been performed taking into account the distortion from perfect octahedral symmetry on the Co_{oct} site, which further support the conclusions drawn. The authors have also performed additional calculations to rule out a low-spin Co⁴⁺ state on this site. An extra discussion of the d-orbital energies associated with the Co_{trig} site has been added, which also provides more clarity.

Overall, this is a very high quality manuscript that has been further improved in the revised version. It will be interesting and relevant to a broad readership and I strongly support publication in Nature Communications.

Re.: Manuscript ID: NCOMMS-19-17691A

“Origin of Ising magnetism in $\text{Ca}_3\text{Co}_2\text{O}_6$ unveiled by orbital imaging” by Leedahl *et al.*
Nature Communications.

Dresden, 15 Oct 2019

Please find below our responses to the referees' comments

Sincerely,

Brett Leedahl, Antoine Maignan, and Hao Tjeng for the authors

Report of Referee #1:

I have examined the reply from the authors and the revised manuscript. At this time, the authors have addressed my comments to my satisfaction and I recommend that the paper be accepted for publication in its current form. I also would like to thank the authors for taking the time to address my comments and for expanding the supplementary materials.

Our response to the Report of Referee #1:

Thank you very much for your valuable comments and positive reply.

Report of Referee #2:

The authors didn't quite answer my question about energy resolution. To be more clear, is the 10.8 eV lifetime broadening just a fit to the experimental data or measured independently, or from literature? If it is only a fit to the data, then improving experimental energy resolution might help.

Our response to the Report of Referee #2:

We apologize for the confusion. We know our experimental broadening, which we determined from the elastic peak: 1.4 eV full-width-at-half-maximum (Gaussian). We observe that the experimental spectrum is composed of three peaks that each have a total full-width-at-half-maximum of about 2.6 eV. We then explain this width difference as the result of a convolution of the experimental resolution (Gaussian) with a lifetime broadening of the core hole (Lorentzian). We find the best fit when we used a Lorentzian with 1.8 eV full-width-at-half-maximum.

Lines 238 – 243 have been updated to clarify this:

...the experimental broadening for all spectra was measured to be 1.4 eV (FWHM), as determined by the elastic peak width. However, it should be noted that the lifetime broadening is about 1.8 eV for this experiment (as determined by the remaining Lorentzian broadening unaccounted for by the experimental Gaussian broadening), resulting in a total broadening of 2.6 eV.

To answer the question whether improving experimental energy resolution might help (or not), we would like to provide the following experimental observations:

Previously we have carried out experiments on NiO and CoV₂O₆ with an experimental broadening of 1.4 eV (Si 111 monochromator) and with a higher resolution wherein the experimental broadening is 0.7 eV (Si 311 monochromator). The results are shown below. The elastic peak shows the different experimental broadening of the experiments. However, both the Ni M₁ and Co M₁ spectra illustrate that the effect of the higher resolution is minimal. This is due to the fact that the Ni and Co 3s core-hole life time broadening dominates. So the present resolution that we have used, i.e. 1.4 eV experimental broadening, is more than sufficient to get the information that we want. That is, for our present experiment, resolving power improvements do not allow us to extract any new information, and only has the effect of drastically reducing our photon flux by about 80%.

We would like to thank the Referee for the valuable comments.

Report of Referee #3:

The authors have clearly answered my previous question regarding the orbital moment associated with the d2 orbital. The manuscript has also been further strengthened in response to the comments of the other referees. In particular, additional crystal field calculations have been performed taking into account the distortion from perfect octahedral symmetry on the Co_{oct} site, which further support the conclusions drawn. The authors have also performed additional calculations to rule out a low-spin Co⁴⁺ state on this site. An extra discussion of the d-orbital energies associated with the Co_{trig} site has been added, which also provides more clarity.

Overall, this is a very high quality manuscript that has been further improved in the revised version. It will be interesting and relevant to a broad readership and I strongly support publication in Nature Communications.

Our response to the Report of Referee #3:

Thank you very much for your time and positive reply.